# Neural Approaches to Short-Time Load Forecasting in Power Systems—A Comparative Study

**Stanislaw Osowski [1,2,*], Robert Szmurlo [2], Krzysztof Siwek [2] and Tomasz Ciechulski [1]**

[1] Institute of Electronic Systems, Faculty of Electronics, Military University of Technology, ul. gen. Sylwestra Kaliskiego 2, 00-908 Warsaw, Poland; tomasz.ciechulski@wat.edu.pl

[2] Faculty of Electrical Engineering, Warsaw University of Technology, pl. Politechniki 1, 00-661 Warsaw, Poland; robert.szmurlo@pw.edu.pl (R.S.); krzysztof.siwek@pw.edu.pl (K.S.)

[*] Correspondence: stanislaw.osowski@pw.edu.pl

**Abstract:** Background: The purpose of the paper is to propose different arrangements of neural networks for short-time 24-h load forecasting in Power Systems. Methods: The study discusses and compares different techniques of data processing, applying the feedforward and recurrent neural structures. They include such networks as multilayer perceptron, radial basis function, support vector machine, self-organizing Kohonen networks, deep autoencoder, and recurrent deep LSTM structures. The important point in getting high-quality results is the composition of many solutions in the common ensemble and their fusion to create the final forecast of time series. The paper considers and compares different methods of fusing the individual results into the final forecast, including the averaging, application of independent component analysis, dynamic integration, and wavelet transformation. Results: The numerical experiments have shown a high advantage of using many individual predictors integrated into the ensemble which are responsible for the final forecast. Especially efficient is the application of non-standard wavelet application in the formation of an ensemble, as well as the use of LSTM as the basic prediction unit. The novelty of the paper is the critical comparative analysis of the time series prediction methods applied for load forecasting in the power system. The presented approach may be useful for the users involved in power system operation management.

**Keywords:** recurrent time series prediction; neural networks; ensemble of predictors; load forecasting; power systems; demand-side management

## 1. Introduction

The economics of the power system operation is one of the most important issues in the power industry. Electricity demand forecasting is a central and integral process for planning periodical operations of the system. The accurate hourly electric load forecasting allows the power operators to adjust the electric supply according to the real-time forecasted load and in this way provides the normal operation of electric power systems [1]. In this context, the accurate forecast of the 24-h load profile for the next day is very important, as this knowledge enables the network operator to include or exclude the active operation of some units in the power plants. Thanks to this the reliability of the power supply and delivery system has increased.

Demand response programs in multi-energy systems including heat and gas are also of great interest nowadays [2,3]. These programs allow counteracting the limited capacity of the power system by integrating more demand-side resources. Hourly forecasting of consumers' electricity demand enables the planning of appropriate management actions under the demand response program. In this way, the hourly power forecasting task can be considered an introductory step in this program.

The electricity load pattern is principally a time series. Therefore, the methods of time series analysis and prediction are usually applied. As with any time series, the load pattern can be represented by a linear or nonlinear model supplied by the carefully selected input attributes.

Nowadays, the most successful in this field are the approaches based on artificial intelligence, especially neural networks. Artificial intelligence and machine learning methods applied to the nonlinear types of neural structures have accelerated the progress in designing the prognosis systems, which better reflect the nonlinear character of the electric load patterns. Many different solutions are based on either feedforward networks, like multilayer perceptrons (MLP), radial basis function (RBF), support vector machine for regression (SVR), or on recurrent structures like Elman or long short-term memory (LSTM). Although many different approaches have been proposed in the past, there still exists the need to improve the accuracy of prediction as much as possible.

In paper [4], the use of support vector regression was proposed, which achieved a mean absolute percentage error (MAPE) of about 2% for the dataset of the North American utility. In paper [5], a hybrid form of predicting electricity demand for 3 building clusters was developed. This used a multi-layer perceptron and a random forest and achieved a MAPE value ranging from 2.56% to 8.10% depending on the cluster. In paper [6], a combined forecasting method based on a backpropagation neural network, an adaptive fuzzy inference network, and a difference seasonal autoregressive integrated moving average system was presented, showing its advantages over the three individual methods working separately. In papers [7,8], the autoencoder was used for short-term electric load forecasting and its advantages in the forecasting tasks were demonstrated.

In recent years, deep learning methods have attracted much attention in predicting short time series [9–14]. The main advantage of deep learning is that the deep structure automatically combines the generation of the diagnostic features and their further processing in regression mode to generate the predicted time series. The examples of such an approach are based either on convolutional neural networks [9–11] or recurrent LSTM networks [12–14]. In papers [9–11], a combination of a convolutional neural network and a recurrent LSTM for load forecasting was presented, which achieved a MAPE of 1.40% for the dataset of a city in northern China [9] and a MAPE of 3.96% for the dataset in the Italy-North region [10]. In [12], an LSTM model was presented using the concept of active learning with moving windows to improve the prediction. In [13], the LSTM-based neural network was used in different configurations in combination with a genetic algorithm to build forecast models for short- to medium-term aggregated load forecasts. In [14], the application of LSTM was demonstrated for 24-h and 1-h forecasting of electricity demand in a large power system and a small power region.

To increase the prediction accuracy, many individual solutions are combined into a final prediction (so-called ensemble of predictors). In [15], some methods of data fusion were analyzed, including averaging, principal component analysis, and blind source separation, showing the advantages of the latter method. Another approach based on dynamic local ensemble integration was proposed in [16] for pollution prediction. The discussed methods are mainly based on the application of different types of neural networks and their combination in an ensemble responsible for the final decision.

This paper presents an overview of pragmatic methods developed by the authors over many years that can be used to build power system load prediction models. The first solutions started from single applications of neural networks. Great attention was paid to the development of a suitable structure of the applied network and the definition of the efficient input attributes. The next steps were aimed at assembling some individual solutions into an ensemble responsible for the final prediction. Such a task requires solving the problems of selecting appropriate members of the ensemble and the way of merging the results predicted by each member. The first approaches in this area relied on the application of neural feedforward predictors based on MLP, RBF, and SVR [15]. The next solutions were based on recently defined recurrent LSTM networks, which are well suited

to the task of time series prediction. To achieve good ensemble performance, special integration techniques were developed. They started with simple weighted averaging of individual results and evolved to more complex strategies based on the use of principal component analysis, independent component analysis, or local dynamic approach [15,16]. In this work, we also propose another philosophy to create the ensemble based on the wavelet transform, which has proven to be very efficient.

The forecasting methods proposed in the paper allow for an increase in the accuracy of time series prediction. The results of numerical experiments performed on the real task in the Polish Power System have shown the possibility of reducing the values of MAPE in a significant way in comparison to the results presented in the papers [1–7].

Any predicting task involves two basic steps: definition of a proper set of diagnostic features of the modeled process and application of them as the input attributes to the regression units. In the classical neural network approach, both steps are separated. The user is responsible for defining the features, which are then delivered to the input of the final regression units, formed usually by neural structures, like MLP, RBF, SVR, Kohonen network, or Elman. The diagnostic features may be selected directly from the measured data or generated algorithmically using, for example, deep structures like autoencoder or CNN. Very interesting is the application of deep LSTM networks, which combine in their structure both tasks inseparably.

Irrespective of the type of the individual predictors they may be combined into an ensemble. The predicted results of the individual ensemble members are fused in the final verdict, which is of better quality (usually better than the best individual result). The main problem is to develop an efficient method of integrating the individual results into the final forecast of the ensemble. The existing approaches, based mainly on averaging, are not very efficient and need some additional study. Therefore, different forms of integration will be presented and compared in this paper. They include weighted averaging of the individual results, application of independent component analysis, wavelet decomposition, as well as the dynamic approach to the integration.

The results of numerical experiments performed on the data of the Polish Power System (PPS) will illustrate these approaches. The real contribution of the paper is to show and analyze the efficiency of different methods of creating the ensemble of predictors composed of different units. The results of experiments have shown a high advantage of using many individual predictors integrated into the ensemble. Different ways of fusing individual predictions have been analyzed and compared. Especially efficient is the application of non-standard wavelet application in ensemble creation as well as the application of the LSTM as the basic predicting unit. The novelty of the work is the critical comparative analysis of the time series forecasting methods used in power system load forecasting. The presented forecasting systems can be useful for users involved in the management of power system operations.

The rest of the paper is structured as follows: Section 2 presents the numerical database of the Polish Power System used in experiments. Section 3 is devoted to the applied methods of forecasting based on the application of artificial neural networks, belonging to feedforward structure, recurrent LSTM, and the self-organizing Kohonen principle. The problems associated with the creation of an ensemble of predictors are considered in Section 4. Section 5 presents and discusses the results of numerical experiments, comparing the accuracy of different approaches to forecasting the 24-h pattern of load demand. The last is the concluding section, which summarizes and discusses the main results of the paper.

## 2. Materials

The database of the Polish Power System of the last few years [17] will be used in all numerical experiments (The database of the Polish Power System within different years is available from https://www.pse.pl/obszary-dzialalnosci/krajowy-system-elektroenergetyczny/zapotrzebowanie-kse, (accessed on 1 January 2022)). The total demand for

power in the system has changed over the years. The yearly average and maximum values in the form of mean and maximal power demand are presented in Figure 1 for the period 1989–2021. Large differences between these two load patterns are visible. For example, in the year 1992, the mean was 17,730 MW with a maximum of 21,508 MW, while in 2013 the mean was 22,150 MW and the maximal demand 24,848 MW.

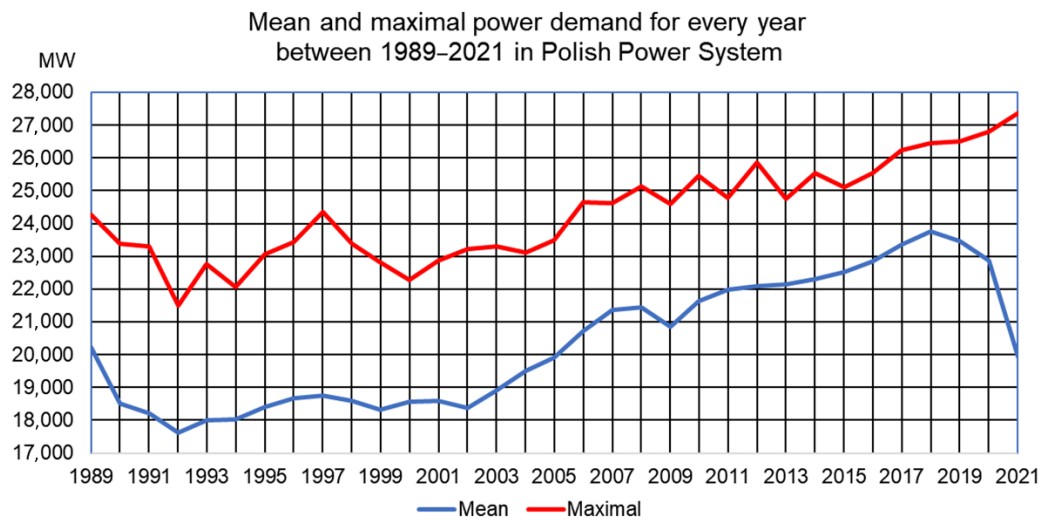

**Figure 1.** Mean and maximal power demands in MW, for every year between 1989 and 2021 in Polish Power System [17].

Predicting the next hour's load demand, we should consider its value from the previous hours. Therefore, the hourly load pattern changes are of great importance. This is illustrated in Figure 2 for the years 2018 and 2021. Irrespective of similar shapes for both years we can observe also great local differences. First of all, the maximum values of power differ a lot. To deal with this the normalization of data for each year is needed. It was done by dividing the real values by their yearly mean. Detailed analysis of the changes in power demand from hour to hour for both years also reveals large differences and no repeatability. The developed model should be resistive to such structural changes. It means that learning data should contain the patterns corresponding to many years (a large learning set).

Moreover, irrespective of the year we can observe some seasonal changes. The highest demand corresponds to the winter season (the first and the last segments of the pattern). The smallest demand for power is observed in the summer months. This observation suggests including this information as additional input to the model. Two bits representing the season, coded in a binary way: winter (11), spring (01), summer (00), and autumn (10), may be added to each partition. The analysis of weekly patterns also reveals significant differences in demand for weekends and the rest of the 5 days of the week. Therefore, the binary code of the working and not working days of the week delivered as additional information to the model, may be of help.

The additional problems in obtaining accurate results of forecasting occur for not typical periods of the year. An example is Christmas and New Year's Eve, in which the load patterns on working and non-working days are different from the typical. The reason is that most institutions and factories have long holidays or work at half-scale. The remedy is to prepare additional network structures trained on the data declared only for this period.

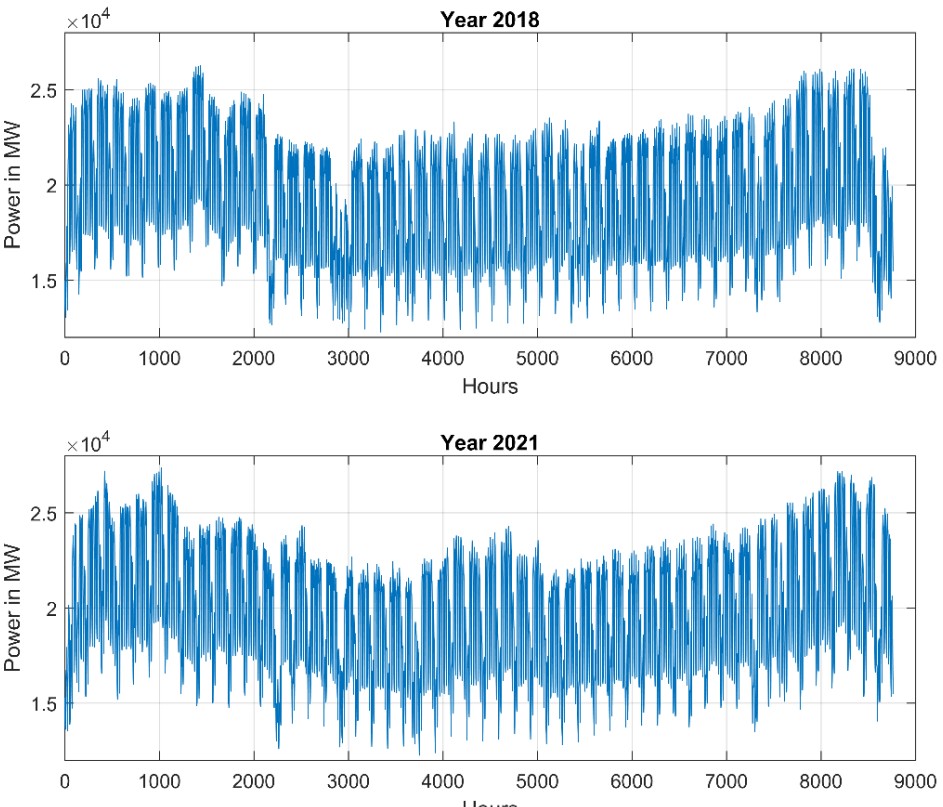

**Figure 2.** The hourly patterns of power demand in the Polish Power System for 2 years: 2018 and 2021.

### 3. Neural Network Structures for Prediction

Many neural network predictors have been developed in the past. The most popular and often used are feedforward structures (MLP, RBF, SVR, Kohonen) and recurrent networks (LSTM). Each may perform the same task and their results might be treated as a final forecast or combined in the ensemble. The important advantage of neural networks is their generalization ability. The structure trained on learning data, after freezing its parameters, performs the role of reproducing the input data to the output signals and such a process is very fast. However, due to the changing trend of power demands within the succeeding years, the system needs additional retraining on the available new data set. According to our experience, it is enough to undertake such retraining every year. The retraining process of the network starts with the current values of its parameters and uses only the newly obtained data samples.

*3.1. Feedforward Neural Predictors*

MLP and RBF neural networks are the universal feedforward approximators. They differ by the applied activation function. MLP belongs to the global approximators since it uses the sigmoidal function for which the neurons participate in the whole range of values of input attributes in generating output predictions. RBF network is the local approximator and operates with Gaussian function representing the local approximation ability. The learning algorithms of both networks are different in a significant way. This is the reason why their output signals in response to the same excitation may also differ and their performance is statistically independent.

Support Vector Regression (SVR) is the support vector machine structure for regression, which can apply any form of a nonlinear kernel satisfying Mercer conditions, although the most universal is the Gaussian kernel [18]. It works in a regression mode, transforming the regression task into classification by defining some tolerance region of the

width ε around the destination. The most important feature of learning tasks is transforming them into a quadratic optimization problem, relatively easy and robust in computer implementation. The learning procedure also uses a few hyperparameters: the regularization constant *C*, the width parameter σ of the Gaussian kernel, and the tolerance *ε*. They should be defined in advance by the user. In practice, their values are adjusted by repeating the learning experiments for the limited set of their predefined values and accepting this one, which results in the minimum error on the validation data set.

All of these networks can share the same set of input attributes, formed from the information related to the past hourly power demands of the system. The analysis of registered load patterns in many years has revealed that hourly changes of load consumption in succeeding hours have a significant dependence on their past values, type of the day (workday or weekends and holidays), and four seasons of the year [15].

Figure 3 depicts a graphical form of the statistical distribution of the power demand depending on the type of the day in the form of a so-called Kohonen map of 49 prototype vectors representing the averaged day load patterns on the weekdays. The 24-h power demand vectors have been grouped in clusters of similar load patterns. The 24-h load patterns corresponding to Saturdays and Sundays create clusters on the right side of the image. The other (working) days from Monday to Friday are grouped on the left and are not interlaced with the weekend days.

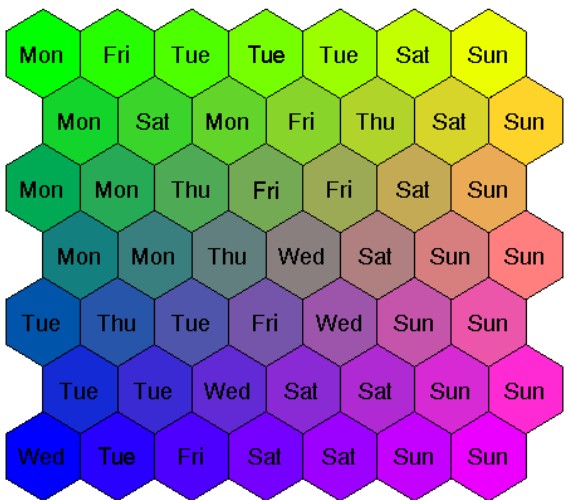

**Figure 3.** The graphical presentation of 24-h load pattern distribution in the form of a Kohonen map corresponds to two types of days: working days (from Monday to Friday) and weekends (Saturdays and Sundays). Both groups of days are well separated.

So, the membership of the day type is the important factor that should be considered in building the mathematical model of the process. As a result, the typical mathematical model of prediction considers the input data in the form of the 24-h load pattern of the previous day, code of the day type (working versus weekend days), and 2-bit code of the seasons of the year (winter, spring, summer, and autumn). Therefore, the input vector to these three neural predictors (MLP, RBF, SVR) responsible for forecasting the 24-h load pattern for (d + 1) day is in the form containing 27 elements,

$$\mathbf{x} = [p(d,1), \dots, p(d,24), \text{season \_code}, \text{day\_code}] \tag{1}$$

in which $p(d,i)$ represents a normalized load of *d*th day and ith hour, the season_code contains 2 bits (11-winter, 10-spring, 00-summer, 01-autumn) and the day_code represents day type (1—working day, 0—non-working day). The destination vector represents the 24-h power pattern for the next (d + 1) day

$$\mathbf{d} = [p(d + 1,1), \dots, p(d + 1,24)] \tag{2}$$

In the case of SVR, only one-hour load demand can be predicted. Therefore, 24 SVR networks responsible for the prediction of the power needed for the succeeding 24 h are needed. Each network is supplied by the same input vector of the form (1).

The other way of creating the input attributes to these predictors is an application of an autoencoder [19]. Autoencoder is a deep neural solution that uses a few hidden layers. The role of succeeding layers is to reduce the dimensions of the input data, step by step. In the learning process, the input data **x** is coded to the vector **h** = **f**(**x**) of reduced dimension. The parameters of the network are adapted in a way to minimize the difference between the actual input vector and its representation reconstructed from the coded one (auto-association mode). To increase the generalization ability, the additional terms representing the regularization are also applied. They consider minimization of weights, the sensitivity of hidden neurons to minor changes in input signal values, as well as specialization of neurons in particular areas of input data [19]. The learning procedure is repeated individually for the succeeding hidden layers. After then the reconstructing parts are eliminated from the structure. The signals of the reduced dimensionality from the last hidden layer are treated as diagnostic features and applied as the input attributes to neural predictors.

Input data to the autoencoder applied in this paper is created from the hourly data of the whole previous week, including the 2-element code of the season, so it is composed of a 170-element vector representing the data of the previous week. The diagnostic features of the reduced dimension are generated from this 170-element vector by an autoencoder as the signals of the last hidden layer. Based on these features delivered to the input of the neural predictor the power forecast for the next week is created by the network. So, this kind of solution can forecast a 24-h load pattern for all days of the next week.

Note, that in contrast to the previous approach, where the form of input data is defined directly by the user, the data characterization formed by an autoencoder is generated by an automatic self-organizing process without human intervention. The different number of hidden layers and neurons should be tried in experiments to get the best performance of the predicting system. The best generalization ability of the system was achieved in the experiments by using two hidden layers. The number of neurons in these layers was selected also in an experimental way.

### 3.2. Recurrent LSTM Approach to Load Prediction

The most efficient approach to time series forecasting is the application of the long short-term memory recurrent neural network [20,21]. This is due to the fact that the hourly needs of power demand are closely related to the previous values, and this is naturally embedded by the nature of the recurrent structure of LSTM [20]. The network structure is composed of an input signal layer, one hidden layer of mutual feedback between neurons represented by so-called LSTM cells, and an output layer, which is fed by the cell signals of the hidden layer. The network is composed of many parallel LSTM cells. The important role of signal processing in the cells performs the so-called multiplication gates, responsible for transmitting the previous information to the next instant of the time. The neuron hidden states pass through time; therefore, the recurrent network can take a learning input sequence of any length up to $t \rightarrow T$ and then can generate an output sequence also of any length. The user decides what is the length of the learning input sequence and what is the length of the predicted output time series values.

The parameters of the LSTM units are adapted in a supervised mode using the set of training sequences and applying a gradient descent algorithm, for example, stochastic gradient descent (SGD) or its modification in the form of adaptive moment estimation (ADAM). The input signals are delivered to the network, pass through the hidden layer and an output signal is calculated. The error between the actual output and the destination signals is backpropagated through the network allowing for the calculation of the gradient, in the gradient descent minimization procedure.

After the training the parameters of the network are frozen and the network responds to the delivered excitation $[x_{t1}, x_{t2},..., x_T]$ and delivers the forecasted next $k$ time point samples representing the future.

### 3.3. Competitive Self-Organization for Load Prediction

Self-organization of data is another approach to time series prediction. The 24-h load patterns are represented by the vectors. These vectors are grouped into a set of clusters according to their similarity measures. The clusters are represented by their centers. To obtain the proper representation of data belonging to the same-day type of previous years trend elimination is needed. In such a case, the 24-h real power consumption vector $\mathbf{P}_j$ corresponding to $j$th day should be normalized to the so-called profile vector $\mathbf{p}_j$, [15]

$$\mathbf{p}_j = \frac{\mathbf{P}_j - \mathbf{P}_m(j)}{\sigma_j} \tag{3}$$

where $\mathbf{P}_m(j)$ represents the mean value vector of real power consumption in $j$th day and $\sigma_j$ its standard deviation. Both parameters are estimated based on data corresponding to the previous years existing in the database. The self-organization procedure using the Kohonen algorithm applied to the set of normalized learning data (profile vectors) leads to the creation of cluster centers representing the set of patterns of 24-h daily load. Figure 4 presents some exemplary distribution maps of 49 center vectors created in the learning process of the Kohonen network. The curve inside of the box depicts the shape of the 24-h load pattern of this particular center.

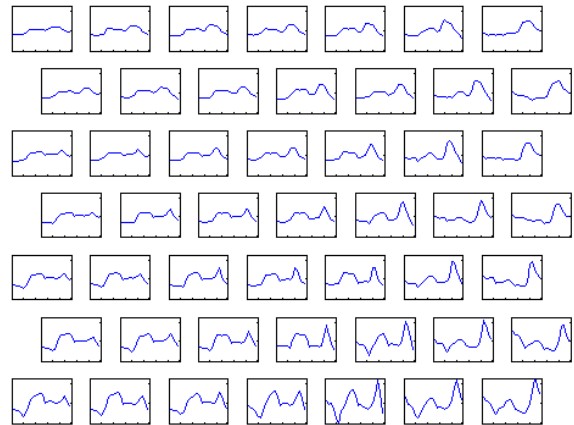

**Figure 4.** The exemplary patterns of the load profiles in the power system generated by the Kohonen network at the application of 49 neurons. Each neuron is represented by the 24 weights corresponding to the center vector of the cluster.

The load demand for any day of the year (for example Tuesdays of July) can be reconstructed as the weighted sum of the learned center vectors $\mathbf{w}_i$ of the Kohonen map, to which the 24-h load demand of the mentioned day belonged in the past. The weight value depends on the number of times this day belonged in the past years. The forecasted profile vector $\mathbf{p}_j$ for $j$th day (for example Tuesdays of July) can be now defined in the form

$$\mathbf{p}_j = \frac{\sum_{i=1}^{n} k_{ji}\mathbf{w}_i}{\sum_{i=1}^{n} k_{ji}} \tag{4}$$

where $\mathbf{w}_i$ is the vector of the $i$th cluster center ($i$ = 1, 2, ..., $n$) and $k_{ji}$ represents the number of times the $i$th center was encountered in the past by the $j$th day under consideration. The value $k_{ji}$ is zero when the $i$th center was never the winner for the considered days. Observe

that this algorithm of profile prediction is applicable for any days ahead. This is a great advantage concerning supervised learning approaches, in which a previous day's load is needed to predict the next one.

However, to obtain the real load vector $\mathbf{P}_j$ of $j$th day we need to predict also the mean and standard deviation for this day, since $\mathbf{P}_j = \sigma_j \mathbf{p}_j + \mathbf{P}_m(j)$. These variables may be estimated using the data nearest to the day for which the forecast is needed or building additional predictors for them.

## 4. Ensemble of Neural Predictors

It was found that the combination of many methods applied simultaneously and integrated into the ensemble outperforms, on average, the individual-specific methods and provides better accuracy of prediction [22]. Ensembles of predictors are regarded now as the most competitive form in predictive tasks. However, the independence of its members is the most crucial condition for success. This may be provided in different ways, for example applying the bagging with different random bootstrap samples of the original training set or using different types and diversified parameters of the predicting units, for example, MLP, RBF, SVR, self-organization, autoregression. Both approaches are used in numerical experiments.

### 4.1. Weighted Averaging

The important point in the ensemble approach is providing the efficient integration (fusion) of the results of its members. In the case of a regression problem, the most often used is weighted averaging, with the weights dependent on the prediction accuracy estimated for each member based on learning results [15,22,23]. In calculating the weights, different approaches might be used: relative accuracy of predictors in the learning stage or application of a special linear combiner. The simplest approach to averaging is taking an ordinary mean of all results. However, this method of fusion may produce final statistical results inferior to the best unit in an ensemble.

### 4.2. Application of PCA and ICA in the Fusion Procedure

The interesting approach to the fusion of many results is to apply the principal component analysis (PCA). The vectors predicted by all units of the ensemble are concatenated into one longer vector, which is subject to PCA decomposition [15]. The limited number of principal components is used in the reconstruction of the original vectors corresponding to all members of the ensemble. In this way, the reconstructed vectors are deprived of the least important components, representing the noise. Averaging these reconstructed vectors generates the final forecast.

The other approach to fusion is a separation of the time series predicted by different units into the set of independent time series and the elimination of terms corresponding to the identified noise. This can be done by the independent component analysis (ICA) [24], delivering the set of independent time series. The process of deflation using only the important components allows for reconstructing the predicted time series deprived of the not important components, treated as the noise [15,24]. This procedure is illustrated in Figure 5. The set of $M$ predictors generates $M$ time series $\mathbf{x}(k)$, which are delivered to the ICA process represented by matrix $\mathbf{W}$. The output signal vector $\mathbf{y} = [y_1(k), y_2(k), \ldots, y_M(k)]^\mathrm{T}$ is the result of linear ICA operation $\mathbf{y} = \mathbf{Wx}$ and contains the independent signals. The switches in the figure represent the possible elimination of the inappropriate independent components at the reconstruction stage of the data. The inversed matrix $\mathbf{W}^{-1}$ represents the deflation process, i.e., reconstruction of the signals.

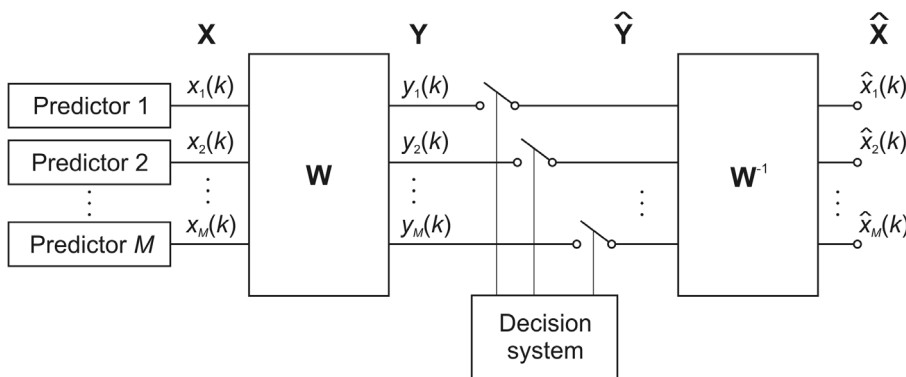

**Figure 5.** The illustration of ICA in the integration process of the ensemble of predictors [15]. The time series predicted by *M* predictors are separated into independent components using ICA represented here by the matrix **W**. The switches are used to eliminate the inappropriate independent components in the reconstruction stage of the data. The passed components are used to reconstruct the noise-free prognosis through matrix **W**$^{-1}$ (so-called deflation process).

The original data **x** is reconstructed by using the inverse operation, called deflation [24]

$$\hat{\mathbf{x}} = \mathbf{W}^{-1}\hat{\mathbf{y}} \tag{5}$$

The variable $\hat{\mathbf{x}}$ denotes the reconstructed time series and $\hat{\mathbf{y}}$ —the independent component data set. The set $\hat{\mathbf{y}}$ is chosen from the original data **y** by eliminating the rows recognized as the noise. If there is a problem with recognizing the noise, we may try all combinations of independent components, substituting the eliminated components (appropriate rows of **y**) with zeros. The best combination of signals obtained in the learning stage represents the final solution. In the testing phase, only this combination is used.

*4.3. Wavelet Application*

The interesting approach to the ensemble prognosis is the application of wavelet analysis [25]. The wavelet decomposition of the load pattern is performed on a few levels. As a result, the analyzed signal is decomposed into as many time series as is the number of levels. Some levels, usually of the highest resolution (the most difficult for prediction), may be treated as noise and eliminated from further considerations. The prediction task of the load pattern is now decomposed into separate predictions of the time series on each decomposition level. This is a much easier task, due to their lower variability compared to the original time series. The prediction is done by building as many neural predictors, as is the number of considered decomposition levels. The predicted signals create a set of partial representations of this load pattern. The reconstruction of the final load values is performed by simply summing the predicted wavelet coefficients.

Figure 6 illustrates the result of wavelet decomposition of the time series representing the hourly load of one year. Daubechies wavelet function db4 [25,26] and five-level decomposition have been applied. The upper curve *s* represents the original time series, $a_5$ is the coarse approximation of the signal *s* on the 5th level, and $d_1$–$d_5$ represents the detailed wavelet representations of the original signal on five levels with different resolutions. The time series $d_1$ is of the highest resolution and includes the lowest impact on the approximation of the original time series. In practice, it may be eliminated from the reconstruction.

The forecasting task is applied for the wavelet coefficients on each level of decomposition including the coarse approximation. In predicting the wavelet coefficients corresponding to *d*th day and *h*th hour the wavelet pattern of the previous day (*d*−1) and the same day one week ago (*d*−7) corresponding to the hours *h*th, (*h*−1), and (*h*−2) create the input attributes. They are supplemented by the day number of the week (the numeric

notation of the day, from 1 to 7) and the numeric notation of the month (from 1 to 12). All data in the columns are normalized linearly to the range from 0 to 1. Any type of neural network or any other predictor can be applied in this process.

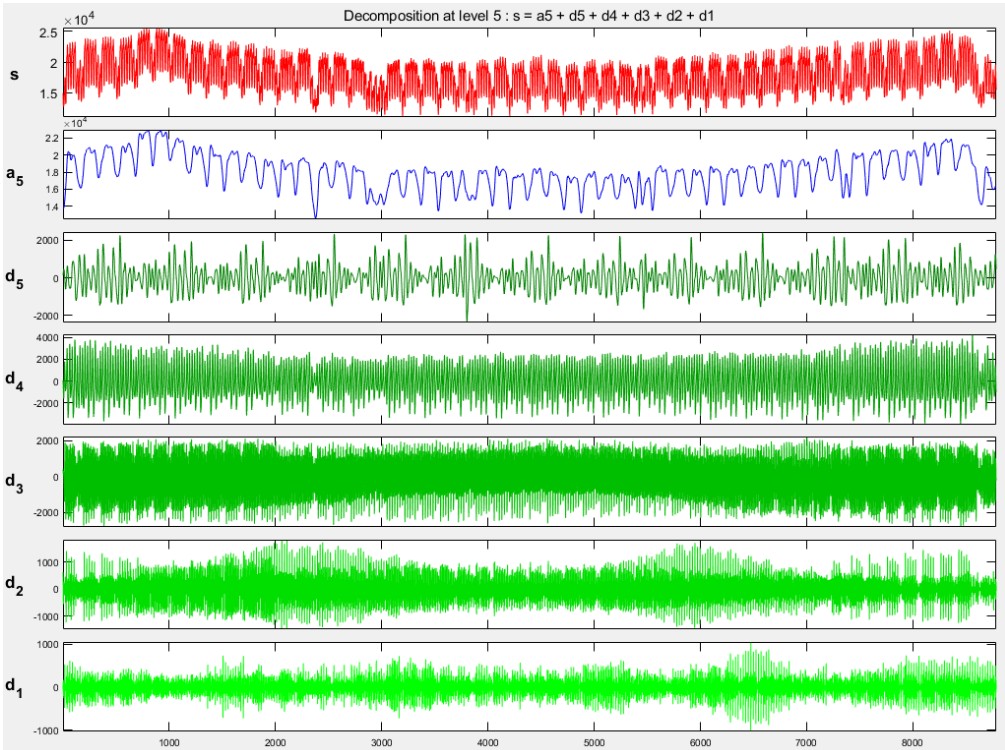

**Figure 6.** The graphical results of wavelet decomposition of the time series represent the hourly load of one year. The upper curve represents the original time series *s*, $a_5$ -the coarse approximation on 5th level, and $d_1$ - $d_5$ - the detailed wavelet representation of the analyzed signal *s* on five levels.

Figure 7 presents the general structure used for the prediction of the wavelet coefficients $D_i(d,h)$ for the particular hour *h* of the day *d* on the *i*th level for *i* = 1, 2,..., *N*. An identical structure is used for the prediction of the coarse approximation signal $A_N(d,h)$.

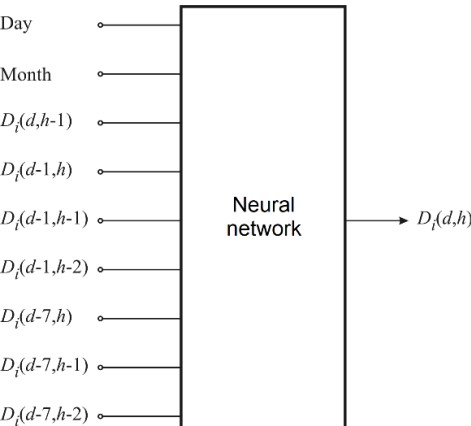

**Figure 7.** The neural system is used for the prediction of wavelet coefficients $D_i(d,h)$. An identical structure is applied for the prediction of the coarse approximation signal $A_N(d,h)$ on the final *N*th level of the wavelet decomposition.

The introductory experiments have shown that five-level wavelet decomposition (*N* = 5) is enough to predict the load pattern for each hour of the day. In this way 6 neural predictive networks (5 detailed coefficient $D_i$ and one approximate (coarse) signal $A_5$ are

used). In the learning process, the destination (the target output signal) is associated with the known value of the appropriate wavelet coefficient for each *h*th hour of the *d*th day. Based on the predicted wavelet coefficients the real prediction of the load at *h*th hour for *d*th day is made by simply adding them (*N* is the number of decomposition levels).

$$s(n) = d_1(n) + d_2(n) + ... + d_N(n) + a_N(n) \tag{6}$$

### 4.4. Local Dynamic Integration

The other interesting approach to the integration of the ensemble is the local dynamic method. The prediction of the time series for the next day is made here by only one member of an ensemble, which was the best in the learning stage for the input vector, closest to the applied input (testing) data. Thanks to such an arrangement we avoid the situation when the worst unit reduces the accuracy of the whole ensemble. The best predictor is selected based on its prediction accuracy for the learning sample in the neighborhood of the actual testing sample. The quality of each member of the ensemble is checked on the learning data closest to the actual testing sample. The most competent predictor, providing the smallest prediction error in the learning mode is chosen. Thanks to this we can get an increased level of forecasting accuracy since each task is performed by the predictor best suited to it. Moreover, such an arrangement of integration allows the use of units of very different statistical accuracy without decreasing the quality of the final prediction.

Given an input vector $\mathbf{x}_t$ in the testing mode, we select its closest neighbor $\mathbf{x}_l$ among all input vectors existing in the learning set. The Manhattan distance measure is used

$$d(\mathbf{x}_t, \mathbf{x}_l) = \left\| \mathbf{x}_t - \mathbf{x}_l \right\|_1 \tag{7}$$

In the next step, we compare the selected quality measure (MAPE, MAE, or RMSE) of the units forming the ensemble, in the regression task for this vector $\mathbf{x}_l$. The member of the smallest prediction error corresponding to $\mathbf{x}_l$ is chosen and applied in the prediction task at the application of $\mathbf{x}_t$. Its generated result is regarded as the final verdict of the whole ensemble. In the case of predicting the time series, each element of this vector might happen to be predicted by different units of an ensemble. Occasionally, two or more predictors might show the same highest local accuracy for the tested vector $\mathbf{x}_l$. In such a case all of them are used in the prediction task. The final decision of an ensemble is their average. In summary, the general organization of the succeeding steps of the proposed signal processing in the forecasting system can be presented in the form of the flowchart shown in Figure 8.

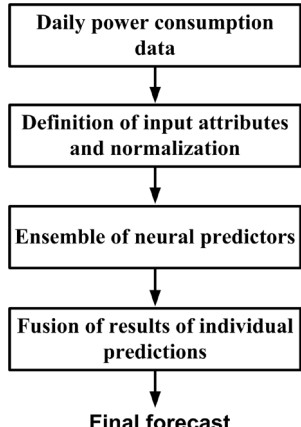

**Figure 8.** The flowchart of signal processing leads to the final forecast of the load pattern.

The input data consists of the daily electricity consumption in combination with information about the type of day and season. The next step is the definition of input

attributes and their normalization. The attributes can be defined manually using expert knowledge or automatically by applying a deep autoencoder. The attributes after normalization are delivered to the set of neural predictors that form the ensemble. The final step is the fusion of the results generated by each member of the ensemble. This general form of the flowchart allows applying different structures of individual predictors as well as various methods of integrating the results of their prediction.

## 5. Results of Numerical Experiments

The numerical experiments have been conducted on the database of the Polish Power System corresponding to many years [17]. They have been performed using the computer with an Intel i7 processor of 32 GB memory and Nvidia GeForce GTX 1080Ti. The computations have been done using the MATLAB platform. The MLP network was trained using Levenberg-Marquardt quasi-Newton method. In training the RBF network, the clusterization combined with the singular value decomposition was used. The support vector machine learning procedure was accomplished by applying a sequential minimal optimization procedure. In the case of the LSTM, the stochastic gradient descent in the ADAM version was applied in learning.

This section will present the results corresponding to the individual solutions and their integration into an ensemble. The comparison will be based on mean absolute percentage error, which is regarded as the most objective measure of prediction accuracy. If we denote by $P(h)$ and $\widehat{P}(h)$ the real and predicted load at $h$th hour, respectively, and by $n$ the total number of hours of prediction, the MAPE is defined as follows [15,23]

$$MAPE = \frac{1}{n}\sum_{h=1}^{n}\frac{\left|P(h)-\widehat{P}(h)\right|}{P(h)}\cdot 100\% \tag{8}$$

This measure may be calculated separately for the learning and testing data. Here we will present only the testing results, related to the data not taking part in learning since this information is the most important from the practical point of view. Moreover, the presented results correspond to the average of many learning/testing runs at different contents of both subsets (each time selected randomly¨ at the proportion of 70% learning and 30% testing).

### 5.1. Individual Predictors

Five years of PPS data have been used in experiments. The learning and testing data have been randomly chosen from this set in the proportion of 70:30. Ten repetitions of experiments at different contents of learning and testing data have been done. The forecast of 24-h load patterns with the application of individual predictors can be solved in different ways. The first results will correspond to the model represented by Equation (1) at the application of MLP, RBF, and SVR of the Gaussian kernel as the predictive units. In this case, the input attributes of the predictors have been selected manually by the user. They were formed by the normalized loads of the nearest past 4 h of the actual day and 5 h (the actual hour and 4 nearest past) for 3 previous days (19 components together). They have been supplemented by the type of season (two nodes coded in a binary way: 00—spring, 01—summer, 10—autumn, and 11—winter) and type of the day (two nodes: 1—working days, 0—non-working days).

The first task in the learning process is to set the correct structure of the neural predictors. This was done utilizing introductory experiments in which different values of the network parameters (number of hidden neurons in MLP and RBF as well as different values of regularization parameter C, coefficient $\sigma$ of Gaussian kernel, and tolerance $\varepsilon$ in SVR) were tried. The additional annual data, which were not used in the further experiments, was used in this stage. The parameter values corresponding to the minimum validation error were applied next. Their optimal values vary according to the different arrangements of the input attributes resulting from the applied definition. The results of

experiments for testing data in the form of the mean value of MAPE in 10 trials and stand-ard deviation are presented in Table 1. As it is seen the MAPE values corresponding to different predictors are close to each other, concerning mean and standard deviation.

**Table 1.** The statistical results of prediction of 24-h load pattern at an expert selection of input at-tributes (number of applied hidden neurons in MLP was 35, in RBF 70, $\sigma = 0.9$, $\varepsilon = 0.01$, and C = 1000 in SVR). They correspond to the testing data. The values of MAPE and standard deviation corre-spond to the mean of 10 repetitions of experiments. Irrespective of the applied neural predictors, their values are not very far from each other.

| Neural Network | MAPE [%] |
|:---:|:---:|
| MLP | 2.08 ± 0.14 |
| RBF | 2.26 ± 0.16 |
| SVR | 2.27 ± 0.08 |

The other approach to input attribute selection for the neural predictors is to apply an automatic system in the form of the autoencoder. The input data to the autoencoder is composed of 170-element vectors representing the load data of the previous week (168 samples) and the 2-element binary code of the season. The autoencoder of two hidden layers was selected. It is responsible for reducing the size of the population, guaranteeing the best performance of the system. The number of neurons in these layers was selected by additional experiments. The last layer of the autoencoder represents diagnostic fea-tures that will be used as the input attributes for the applied neural predictors: MLP, RBF, and SVR. Note, that in contrast to the previous approach, where the input features of data were chosen by the expert, the input vector to the above predictors in this method is gen-erated by an automatic self-organizing process without human experience-based inter-vention. The solution has been applied in forecasting the 24-h power demand for 7 days of the next week. Despite the increased horizon of prediction, the numerical results were quite encouraging, however, different for each type of predictor. The best testing results of each predictor have been obtained at different structures of the autoencoder. The details of the autoencoder structure, as well as the numerical values corresponding to the testing results in the form of MAPE and standard deviation for the best choice of autoencoder layers, are presented in Table 2. This time the highest accuracy has been obtained at the application of the RBF network as the predicting unit. The application of SVR was the least efficient (the highest MAPE value).

**Table 2.** Mean values of MAPE testing error for three individual neural predictors cooperating with an autoencoder (number of hidden neurons of MLP was 40, in RBF 50, $\sigma = 0.5$, $\varepsilon = 0.01$, and C = 3000 in SVR).

| Autoencoder Structure | Neural Predictors | | |
|:---:|:---:|:---:|:---:|
| | MLP | RBF | SVR |
| 170-60-30 | 1.96 ± 0.19% | | |
| 170-80-40 | | 1.81 ± 0.14% | |
| 170-95-50 | | | 2.13 ± 0.07% |

The self-organizing (Kohonen) approach to forecasting the profile vectors has been applied using 100 neurons, chosen on the ground of good generalization ability of the network. The mean value $P_m(d)$ and standard deviation $\sigma(d)$ of day $d$, which are needed in the final prediction of the real load demand for $d$th day, have been predicted using the additional MLP network. The number of self-organizing neurons and the structure of MLP have been adjusted after a series of numerical experiments using the validation data [17]. When predicting the profile vector for a particular day of the week (e.g., Monday in July), we estimate it by averaging the winning vectors for that day (e.g., all Mondays in July) from the past learning data using Equation (4). The final MAPE value depends on

the accuracy of the profile estimation and the prediction of the mean and standard deviation. The prediction of standard deviation had the largest effect on this measure since its value was characterized by the largest changes from one day to the next day. The final statistical results for the test data in terms of the mean of MAPE and the standard deviation of a total load of days were MAPE = 2.34 ± 0.16%. This value is slightly worse than the results of the neural predictors shown in Table 1. However, this approach can be applied to any day of the year at any time in advance.

The LSTM predicting system relies on a sufficiently different principle since it takes directly into account the memorized previous load pattern. The input data delivered to the network is composed of a series of 24-dimensional vectors **x** representing the 24-h patterns of the previous daily load. They are associated in the learning stage with the predicted target vector of the 24-h load pattern of the next day. The training procedure uses the pairs of vectors: input $\mathbf{x}(d)$ and output $\mathbf{x}(d+1)$. The testing phase is very simple. The load pattern for the next day is predicted by the trained network based on the delivered input vector of the already known pattern of the previous day.

In the experiments, we used the structure of the LSTM network 24-$n_h$-24, where $n_h$ represents the number of LSTM cells. In the experiments, the input pattern to the network is composed of 24 h loads of the last (known) day. As a result of introductory experiments, the number of hidden neurons was set to 600.

In the PPS we have observed high differences in patterns in the period of the last 10 days of the year, including the Christmas holiday and New Year's Eve. The load pattern in this period is different from day to day and is not repeatable. The reason for this is that during this period most institutions and factories have long holidays or work on half-scale on other days.

Since the operation of LSTM is sensitive to the shape of learning patterns, we have decided to learn two different LSTM models. In the case of a network designed for the prognosis of the Christmas holiday period, the population of data is very small. Therefore, the experiments have been done including together the data of the last 10 days of the six years (from 2014 to 2019) resulting in six runs of learning and testing phases. Five-year data (the combinations of five years of data in the period 2014–2019) have been used in learning and the remaining year for testing. The experiments have been repeated 6 times for each arrangement of learning and testing data. The mean of all testing results represents the forecast for this holiday period [14].

In the second case (all days of the year except the last 10 related to Christmas and New Year's Eve) the experiments have been organized differently. The data of 3 years (2017, 2018, and 2019) have taken part in experiments. One-year data (355 days of it) was used for learning and the whole next year's data were served for testing. In this way, three independent LSTM systems of the same architecture and hyperparameters have been learned. The first was trained on the data set of 2017 and predicted the daily load for the year 2018. The second was trained on the data of 2018 and predicted the daily load of the year 2019. The third system for predicting the load in 2017 used the data set of 2018 in training. The final statistical results of these experiments are presented in Table 3 [14]. They represent the average values and standard deviations of MAPE obtained in all runs of experiments.

**Table 3.** Statistical results of numerical experiments in 24-h load pattern prediction in PPS using LSTM predictor.

| Year | MAPE [%] |
|---|---|
| Days except for the last 10 days of the year | 1.52 ± 0.09 |
| Last 10 days of the year | 3.53 ± 0.70 |
| Weighted average for the whole year | 1.63 |

As it is seen LSTM prediction results represent the highest accuracy. This is well seen in the case of a typical day, where the MAPE error achieved a very small value. However, in the case of the not typical period (like the Christmas holiday), the performance of the LSTM predictor is significantly worse. This is mainly due to the very small amount of learning data available for this short 10-days Christmas period.

### 5.2. Ensemble of Predictors

The ensemble of predictors is composed of many independent units operating on the same database. To provide the independence of predictors different approaches are used: bagging of the learning data from the learning set, different types of predictors, and different values of hyperparameters. The other crucial point is the fusion procedure. The results predicted by the members of the ensemble are combined into a final forecast. Here we will compare different solutions concerning the MAPE value calculated for the same, common sets of testing data.

The simplest approach to the fusion of results is averaging the predictions of all members of the ensemble. Different arrangements of predictors can be tried. In the case of MLP, RBF, and SVR predictors supplied by the input attributes chosen manually according to the expert's knowledge, the final value of MAPE was reduced from an average of 2.11% (Table 1) to 1.83%. The application of PCA with 11 principal components to the transformation of individual predictions into diagnostic features for these 3 neural networks has reduced the MAPE to 1.81%. Replacing the PCA procedure with ICA has further reduced this value to 1.73%.

Better results of averaging have been obtained with the application of autoencoder in the role of automatic generation of the input attributes for these 3 neural networks. The MAPE value has been reduced to 1.61%. Application of dynamic fusion was found not competitive. The MAPE, in this case, was reduced to only 2.09% from the average of 2.11%.

Interesting results have been obtained with the application of wavelet transformation. The time series representing the hourly load pattern of each year has been decomposed onto the detailed coefficients of 6 levels and the approximated signals on the 6th level. All of them have been transformed to the standard resolution. Half of the data of each year has been chosen for learning and the second half for testing purposes. The division was done randomly in such a way that each month of the year has been represented in both sets. Two neural networks (MLP and SVR) have been trained for the prediction of the wavelet coefficients of each level. In this way, 7 predictive networks specialized for the prediction of 6 detailed coefficients (for each of 6 levels) and one network for the approximated signal on the 6th level have been trained. The trained networks have been tested on the testing set, not used in learning. The resulting MAPE at the application of MLP as the applied predictors was 1.98%, while at SVR application the MAPE was much smaller and equal to 1.54%.

The principle of application of LSTM in the predictive model is sufficiently different from previous approaches and could not cooperate directly with them in the ensemble system. However, it is possible to arrange the ensemble composed of only LSTM networks, by applying many runs at different hyperparameter values of the networks and integrating their results. The MAPE value of the ensemble is integrated in the following way

$$MAPE_{ensemble} = \frac{1}{p}\sum_{h=1}^{p}\frac{\left|y_m(h)-d(h)\right|}{d(h)} \cdot 100\% \qquad (9)$$

where $p$ represents the number of hours for which the predictions are made, $y_m(i)$ is the mean of values predicted for $h$th hour by all members of the ensemble, and $d(h)$ is the true value of the load at the $h$th hour. The number of repetitions of the learning/testing procedure represents the number of ensemble members. The experiments have shown that 5 members are optimal. Their application has reduced the mean value of MAPE from 1.52%

(Table 3) to 1.46% (for the data except for the Christmas period) and for all days (including Christmas) from 1.63 to 1.53%.

Figure 9 shows the graph of MAPE values corresponding to different approaches to load forecasting. Their positions, denoted by the following numbers, represent:

1. individual MLP predictor at the manual expert selection of input attributes
2. individual RBF predictor at the manual expert selection of input attributes
3. individual SVR predictor at the manual expert selection of input attributes
4. individual MLP predictor employing autoencoder for selection of input attributes
5. individual RBF predictor employing autoencoder for selection of input attributes
6. individual SVR predictor employing autoencoder for selection of input attributes
7. self-organizing Kohonen network application
8. individual LSTM predictor
9. ensemble composed of MLP, RBF, and SVR integrated by ordinary averaging for succeeding hours
10. ensemble composed of MLP, RBF, and SVR integrated using PCA
11. ensemble composed of MLP, RBF, and SVR integrated using ICA
12. ensemble composed of MLP, RBF, and SVR integrated using autoencoder
13. ensemble based on wavelet decomposition and application of SVR
14. ensemble composed of 5 integrated LSTM predictors

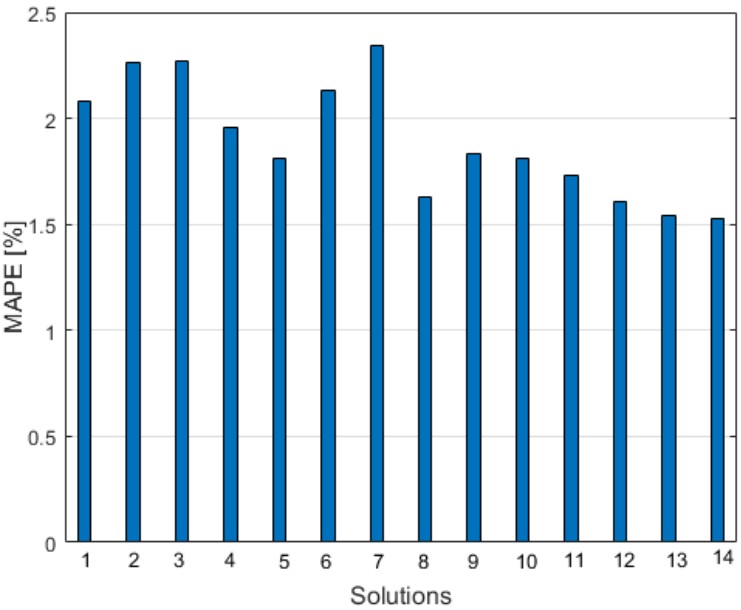

**Figure 9.** The comparison of MAPE values at the application of different individual tools as predictive units and their integration in the ensemble. The particular solutions are denoted by numbers from 1 to 14. LSTM is unbeatable in all cases.

As is seen, the best results correspond to the application of the LSTM predictor, both in the individual role and also in an ensemble arrangement. It is also seen that in all cases the ensemble of predictors is superior to the individual performance. However, the relative improvement of results differs and depends on the applied strategy of fusion. In the case of the application of feedforward neural networks (MLP, RBF, SVR) the rate of improvement will be calculated concerning the mean of their average individual performance (MAPE = 2.20%). In the case of LSTM, this rate is compared to the individual performance of the LSTM predictor (MAPE = 1.63%). Figure 10 depicts such a graph.

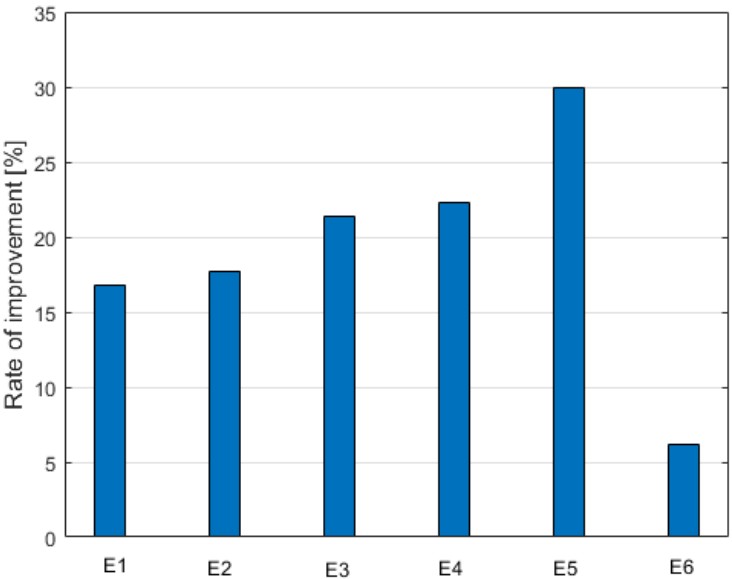

**Figure 10.** The relative improvement of MAPE at the application of the different methods in the fusion of individual results of ensemble members. The first 5 bars represent the feedforward solution of predictors and the last corresponds to the recurrent LSTM network.

The following bars represent the percentage value of improvement related to the following methods of integration

E1—integration by averaging results of MLP, RBF, and SVR for particular hours
E2—application of PCA in integration
E3—application of ICA in integration
E4—application of autoencoder in the integration scheme of MLP, RBF, and SVR
E5—wavelet approach combined with SVR compared to the classical application of SVR
E6—integration of a few LSTM predictors compared to individual LSTM performance

The relative improvement of the best predictor (LSTM) is the lowest one. This is because the application of the same type of predicting units changing only by the limited number of hyperparameters and changing contents of the learning data does present only the limited independence of its members. Moreover, the reference point of the accuracy of the individual members was of a very high level. Relatively large is the improvement introduced by the wavelet transformation. Observe, that in this case, the wavelet terms on different decomposition levels cooperating with the SVR predictor represent different approaches to the methodology of building the ensemble system. Therefore, their independent operation is achieved to a very high degree.

## 6. Conclusions

The paper has presented a comparative analysis of the performance of different neural-based approaches to predicting the hourly electricity demand (24-h load patterns) in electrical power systems. Two types of approaches have been presented: the individual solutions based on a choice of a single neural predictor and an ensemble of many predictors combined into one final forecast obtained by integration of a few individual predictions. The considered neural network solutions include feedforward supervised structures (MLP, RBF, and SVR), recurrent networks (LSTM), and self-organizing Kohonen networks. Different approaches to creating the set of input attributes to the networks have been presented: expert suggested choice, application of additional analysis of data based on PCA, ICA, wavelet transformation, or deep autoencoder.

The accuracy of each predictor depends on the type of network structure used. The lowest MAPE value in the test data was observed for the recurrent LSTM network. Despite

a very simple prediction model in this solution, the prediction accuracy obtained with the LSTM is better than when standard feedforward neural predictors are used. This is because the recurrent structure of this network allows reflecting very well the trends of the changing character of the power consumption.

The extensive numerical experiments performed on the PPS database have shown that by combining the ensemble of predictors and the sophisticated input attribute generation system, the accuracy over the individual predictors can be significantly increased. The improvement is made possible by mutual compensation of the errors committed by the individual predictors. The rate of improvement observed for the prediction of the 24-h load profile in PPS, due to the application of the ensemble, varied between 6% and 30%, depending on the type of ensemble created. In this case, the application of the wavelet transforms and the division of the prediction task among the predictions of many decomposition levels is particularly efficient.

All experiments have been carried out for the data of PPS. It is rather difficult to compare these numerical results to the results obtained for power systems of different countries. Comparing the detailed results that correspond to the very different load patterns of countries will not be fair. It is well known that the results of the predictions depend heavily on the complexity of the patterns, which change considerably for different countries. However, the comparative relative results corresponding to different approaches considered in the paper are still of great information, valid also for other world regions. In such a case the analysis of the available data set is required to find the most important factors characterizing the days' patterns of the week and the seasons of the year. It is natural that the data distributions in tropical countries differ a lot from European countries and in input data preparation this fact should be considered.

It should be noted that the accuracy of the proposed systems based on ensemble is on the acceptable level to the experts working in the electricity markets of the country. Therefore, the method has a certain perspective for practical application.

From the research point of view, the presented methods indicate the new directions in developing efficient approaches to load forecasting in power systems. Especially interesting is the application of wavelet decomposition, which was never used in the previous approaches to load forecasting. Moreover, the forecasting methods presented here can be adapted quite easily to other time series forecasting tasks, such as forecasting the demand for different types of fuels. The neural network structures used and the way the ensemble is created provide a universal approach to such problems.

The presented system of the highest accuracy is composed of many predicting units combined into an ensemble. Thanks to such an arrangement increased accuracy in comparison to the existing approaches [1–7] is obtained. A disadvantage of such a solution is the long training time, since each unit should be trained separately. The learning process takes approximately a few minutes for each network. Future work will focus on a parallel approach to the learning procedure of each unit. The other direction is to apply the other deep structures in the time series prediction system and the extension of the ensemble models by using a larger number of different units implementing various prediction mechanisms.

**Author Contributions:** Conceptualization, S.O. and K.S.; methodology, S.O., K.S.; software, T.C., K.S. and S.O.; validation, T.C. and R.S.; formal analysis, K.S. and R.S.; investigation, T.C. and R.S.; resources, K.S. and T.C.; data curation, R.S. and T.C.; writing—original draft preparation, S.O.; writing—review and editing, S.O., R.S. and T.C.; visualization, K.S. and R.S.; supervision, S.O.; project administration, R.S. and S.O.; funding acquisition, R.S. and K.S. All authors have read and agreed to the published version of the manuscript.

**Funding:** This research received no external funding.

**Institutional Review Board Statement:** The study did not require ethical approval.

**Informed Consent Statement:** Not applicable.

**Data Availability Statement:** The database of the Polish Power System within different years is available from "https://www.pse.pl/obszary-dzialalnosci/krajowy-system-elektroener-getyczny/zapotrzebowanie-kse" (accessed on 1 January 2022).

**Conflicts of Interest:** The authors declare no conflict of interest.

**Abbreviations**

The following abbreviations are used in this manuscript:

| | |
|---|---|
| PPS | Polish Power System |
| MLP | Multi-layer perceptron |
| RBF | Radial basis function |
| SVR | Support vector machine in regression mode |
| CNN | Convolutional Neural Network |
| PCA | Principal Component Analysis |
| ICA | Independent Component Analysis |
| LSTM | Long Short-Term Memory |
| MAPE | Mean Absolute Percentage Error |
| SGD | Stochastic Gradient Descent algorithm |
| ADAM | ADAptive Moment estimation algorithm |
| SGD | Stochastic Gradient Descent algorithm |
| ADAM | ADAptive Moment estimation algorithm |

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
