# Peer review of "Neural Approaches to Short-Time Load Forecasting in Power Systems—A Comparative Study"

_energies, doi:10.3390/en15093265_

Round 1

Reviewer 1 Report

The paper presents the techniques of neural network approach to short-time load forecasting in Power Systems. The study discusses different methods of data processing, at the application of feedforward and recurrent neural structures. I think the work merits a publication in this journal, but i have the following concerns:

1) Expectedly, load forecasting directly affect the deployment of demand response (DR) programs, which is not considered here in this paper. There are many DR programs as reviewed in ["Impacts of demand-side management on electrical power systems: A review", Energies] before. The author should explain/discuss, which DR will be of most concern and provide the reason in their paper.

2) Following the above, the DR program with load forecasting has been applied before in various studies. These studies have demonstrated before the application of load forecasting alongside demand response, and shows the interconnectivity between the two considerations, which are lacking in this paper. Hence, I think the authors should additionally consider an appropriate DR program in their existing model, or to provide a qualitative discussion highlighting the drawback of their paper from the perspective of the above work.

Author Response

Response to the reviewer suggestions are in the file Reviewer1.docx

Reviewer 2 Report

In this paper, authors have presented a comparative study of neural network structures as applied to short term load forecasting problem. Although paper is easy to follow, but it needs improvement. I have following suggestions:

  1. There are grammatical mistakes in the paper. Authors should pay efforts to rectify all such errors.
  2. Literature review is weak and should be expanded. This topic is well-investigated in literature and therefore authors should include material in Introduction section.
  3. It is better to include graphical structures of neural networks for the sake of completeness.

Author Response

Response to the reviewer suggestions are in the file Reviewer2.docx

Reviewer 3 Report

I have reviewed the manuscript "Neural Approaches to Short-time Load Forecasting in Power Systems – a comparative study", Manuscript ID: energies-1659951. In this paper, the authors present a comparative analysis of the performance of different neural-based approaches designed to predict the hourly electricity demand (24-hour load patterns) in electrical power systems. The manuscript under review is interesting. However, I consider that the article will benefit if the authors take into account the following remarks and address within the manuscript the signalled issues:

Remark 1: the main strong point of the manuscript consists in the fact that it approaches a very interesting topic for the experts in the field.

Remark 2: the main weak point of the manuscript under review consists in the fact that even if the authors have documented their research, there are still some issues regarding the information provided within the paper. In the actual form of the manuscript, excerpts of the text from Manuscript ID: energies-1659951 are identical to ones from other previously published papers (which are not even cited) namely:

  • Ciechulski, T.; Osowski, S. Deep Learning Approach to Power Demand Forecasting in Polish Power System. Energies2020, 13, 6154. https://doi.org/10.3390/en13226154
  • Osowski S., Siwek K.: Local dynamic integration of ensemble in prediction of time series, Bulletin of the Polish Academy of Sciences, Technical Sciences, PAN, vol. 67, no. 3, 2019, pp. 517-525, DOI:10.24425/bpasts. 2019.129650
  • Siwek, Krzysztof & Osowski, Stanislaw & Szupiluk, Ryszard. (2009). Ensemble Neural Network Approach for Accurate Load Forecasting in a Power System. Applied Mathematics and Computer Science. 19. 303-315. 10.2478/v10006-009-0026-2.

 The authors must address this issue either by writing the information using their own words in the manuscript or by using quotation marks. However, taking into account that the used information does not represent famous quotes, statements of elderly scholars, I find it more suitable for the authors to express the concerned ideas with their own words.

The above-mentioned papers are published by research teams comprising a part of the authors of the manuscript under review. Consequently, the authors have succeeded in establishing a precedence in their line of research. Therefore, I consider that the manuscript under review will benefit a lot if the authors highlight how the timeline of their research has evolved from the findings of their previous studies to the present research results reported in the current manuscript. In my opinion, making a detailed comparison between the current approach along with the results from the paper under review and the ones from the previous papers, is a very important, relevant and mandatory aspect. In this context, I would like the authors of the Manuscript ID: energies-1659951 to highlight clearly, by writing in the paper what are the main differences between their conducted study from the manuscript and the previous published ones.

Another weak point of the manuscript under review consists in the fact that even if the authors have cited a series of papers when presenting their obtained results, these citations are used in order to sustain and justify the statements from the manuscript under review, but not in the purpose of devising a comparison between the authors' obtained results / devised approach and other existing ones from the literature. Therefore, the manuscript does not reflect sufficiently the way in which the obtained approach can be perceived in perspective of previous studies that have tackled similar problems. This comparison is mandatory in order to highlight the clear contribution to the current state of knowledge that the authors have brought.

If the above-mentioned problems are solved, I consider that the paper will benefit if the authors address within the manuscript the following aspects:

Remark 3: Lines 12-20, the "Abstract" of the manuscript. The manuscript will benefit if the authors provide a structured abstract, that covers the following aspects: the background (in which the authors should place the issue that the manuscript addresses in a broad context and highlight the purpose of the study), the methods used to solve the identified issue (that should be briefly described), a summary of the article's main findings followed by the main conclusions or interpretations. In the abstract the authors must also declare and briefly justify the novelty of their work. The authors must present in a clearer manner the above-mentioned aspects: the background, the methods, the main findings, the conclusions, as in the actual form of the manuscript, the abstract offers information related only to some of these aspects and even so, their delimitation is unclear.

Remark 4: the Literature Review. In its current form, the "Introduction" section contains a series of cited papers. I do not contradict the value of these papers, or their relevance in this context, but I consider that the article under review will benefit if the authors extend this section by analyzing appropriately the cited papers. I consider that the literature review should be improved by performing a careful analysis of the cited works. The authors must highlight for the involved referenced papers the main contributions that the authors of the referenced papers have brought to the current state of knowledge, the methods used by the authors of the referenced papers, a brief presentation of the main obtained results and some limitations of the referenced articles. This is the only way to contextualize the current state of the art in which the authors of the manuscript position their paper, identify and address aspects that have not been tackled/solved yet by the existing studies.

Remark 5: the gap from the current state of knowledge that the manuscript intends to fill. After having performed the critical survey of what has been done up to this point in the scientific literature, the authors should identify and state clearer the precise gap in the current state of knowledge that needs to be filled, the gap that is being addressed by their study. This approach will be extremely helpful when discussing the obtained results of the manuscript as well. In the "Introduction" section the authors must also declare the novel aspects of their work. After having declared the novel aspects of their work, at the end of the Introduction, the authors should preview the structure of their paper, under the form: "The rest of the paper is structured as follows: Section 2 contains…"

Remark 6: the details regarding the datasets. It will benefit the manuscript if the authors provide more details regarding the datasets used in their paper. At Lines 74-75, the authors state: "The database of the Polish Power System of the last few years [15] will be used in all numerical experiments.". It will benefit the manuscript if the datasets that the authors have used in their experimental setup are provided as supplementary materials to the manuscript as the authors should provide all the necessary details in order to allow other researchers to verify, reproduce, discuss and extend the obtained published results of the authors. Other researchers should not have to obtain and concatenate the datasets using various methods, risking in obtaining different datasets or datasets that have been normalized differently than the datasets on which the authors have performed their experiments and analysis.

Remark 7: the generalization capability of the developed approach. Can the authors mention how much of their model is being influenced by the used data or to which extent the model can be easily applied to other situations, when the datasets are different? In this way, the authors could highlight more the generalization capability of their approach in order to be able to justify a wider contribution that has been brought to the current state of art.

Remark 8: preprocessing data. I consider that the authors must provide more details regarding the preprocessing approach that they have used and the way in which they intend to solve the problems related to missing data or abnormal values if they are to occur within the datasets. 

Remark 9: the flowchart. I consider that in addition to the actual explanations, in order to help the readers better understand the methodology of the conducted research, the authors should devise a flowchart that depicts the steps that the authors have processed in developing their research and most important of all, the final target. This flowchart will facilitate the understanding of the proposed approach and at the same time will make the article more interesting to the readers if used as a graphical abstract.

Remark 10: the software and the detailed hardware configuration. It will benefit the paper to specify details regarding the version numbers for the software and the detailed hardware configuration used to obtain the results.

Remark 11: issues regarding the equations within the manuscript. All the equations within the manuscript should be explained, demonstrated or cited, as a part of the equations have not been introduced in the literature for the first time by the authors and they are not cited.

Remark 12: the devised approach. At Line 459, the authors state: " Half of the data of each year has been chosen for learning and the second half for testing purposes." The paper will benefit if the authors present more details regarding the results obtained during various tests, for different tested data division ratio values, up to the moment when the chosen ratio has proven to be the best (or suitable) approach and which was the criterion/performance metric used in choosing this ratio.

Remark 13: the neural network approach. As the authors have used a neural network approach, I consider that they must specify in the paper how often does each network need to be retrained/updated and how did they tackle the need of retraining/updating the developed networks.

Remark 14: the retraining process. How is the new data encountered stored for subsequent updates of the network?

Remark 15: the training/retraining times. The paper will benefit if the authors present more details regarding the results obtained during various tests, for all the different number of neurons and epochs tested and especially the training time for each test, until they have obtained the configuration that has provided the best results. The information can be summarized in a table and if it becomes too long, the authors can restrict it in the paper to ten main experimental runs, and a complete table with all the experimental tests can be inserted in the "Supplementary Materials" file of the article.

Remark 16: the training algorithms. It will benefit the paper if the authors extend their paper by providing more details regarding the training algorithms that they have used when developing their Artificial Neural Network approach and the reason(s) that stood behind the decision of choosing these training algorithms.

Remark 17: the advantages and disadvantages of the proposed approach. The authors should underline both the advantages and disadvantages of their proposed approach when compared with other valuable studies from the current state of art. When discussing their obtained results, the authors should emphasize not only the novel aspects and strong points of their developed method, but also to point out objectively the existing limitations of their method, possible circumstances that might hinder their method’s effectiveness and state clear and accurate directions they will pursue in their future research activities in order to extend the current research and overcome these limitations.

Remark 18: discussing the obtained results – insight. The paper will benefit if, after having discussed the obtained results, the authors make a step further, beyond their approach and emphasize more the insight regarding what they consider to be, based on the obtained results, the most important, appropriate and concrete steps that all the involved parties should take in order to benefit from the results of the research conducted within the manuscript. In addition, due to the fact that (as stated at line 31) the authors have obtained, based on their developed approach, the day ahead forecasted load, I consider that it will benefit the paper if the authors explain within it the usefulness of such a forecast for electricity producers that have to provide accurate day-ahead forecasts.

Minor remarks:

  • The "Materials and Methods" section is missing. First of all, according to the "Instructions for Authors" from the Energies MDPI Journal's website (https://www.mdpi.com/journal/energies/instructions), each manuscript must contain a "Materials and Methods" section ("Energies now accepts free format submission: We do not have strict formatting requirements, but all manuscripts must contain the required sections: Author Information, Abstract, Keywords, Introduction, Materials & Methods, Results, Conclusions, Figures and Tables with Captions, Funding Information, Author Contributions, Conflict of Interest and other Ethics Statements."). Therefore, I consider that the authors must devise a proper "Materials and Methods" section. In order to bring a benefit to the manuscript, the authors should mention early in the "Materials and Methods" section, preferably in the first sentence, the choices they have made in their study. The authors should state what has justified using the given method, what is special, unexpected, or different in their approach.
  • Lines 63-64: "neural structures, like MLP, RBF, SVM, Kohonen network, Elman, etc."; Lines 232-233: "for example, MLP, RBF, SVR, self-organization, autoregression, etc."; Lines 238-239: "relative accuracy of predictors in the learning stage, application of learned linear combiner, etc."; Lines 440-441: "bagging of the learning data from the learning set, different types of predictors, different values of hyperparameters, etc.". In a scientific paper one should avoid using run-on expressions, such as "and so forth", "and so on" or "etc.". Therefore, instead of "etc.", the sentences should mention all the elements that are relevant to the manuscript.
  • Lines 14-16: "They include such networks as multilayer perceptron, radial basis function, support vector machine, self-organizing Kohonen networks, deep autoencoder, and recurrent deep LSTM structures." Even if it is widely known in the scientific community, the LSTM acronym, as well as any other acronyms, should be explained the first time when they appear in the manuscript.
  • According to the MDPI Energies Journal's Template, all the figures should be referred in the main text as "Figure X". In the manuscript under review, sometimes the figures' citations appear in the main text as "Fig. X". Please address this issue by modifying the way in which the figures are referred in the main text, according to the Journal's Template.
  • The format of the paper. Regarding the format of the paper, the authors must take into account the Instructions for Authors from the Energies MDPI Journal's website and the recommendations from the journal's Template ( https://www.mdpi.com/journal/energies/instructions ).

Author Response

Response to the reviewer suggestions are in the file Reviewer3.docx

Round 2

Reviewer 3 Report

I have reviewed the revised version of the manuscript "Neural Approaches to Short-time Load Forecasting in Power Systems – a comparative study", Manuscript ID: energies-1659951 and I can state that the manuscript has been improved in contrast to the previous submission.

Author Response

(The authors gave the same response as above.)
